# What is an invasive procedure? A definition to inform study design, evidence synthesis and research tracking

Sian Cousins,[1] Natalie S Blencowe,[1,2] Jane M Blazeby[1,2]

¹Centre for Surgical Research, Population Health Sciences, University of Bristol, Bristol, UK
²Division of Surgery, Head and Neck, University Hospitals Bristol NHS Foundation Trust, Bristol, United Kingdom

**Correspondence to**
Dr Sian Cousins;
sian.cousins@bristol.ac.uk

## ABSTRACT

Worldwide, there are at least 230 million invasive procedures performed annually and most of us will undergo several in our lifetime. There is therefore a need for high-quality evidence to underpin this clinical area. Currently, however, there is no widely accepted definition of an invasive procedure and the terms 'surgery' and 'interventional procedure' are characterised inconsistently. We propose a definition for invasive procedures which addresses the limitations of those currently available. Our definition was developed from an analysis of the 3946 papers from the last decade. A preliminary definition was created based on existing definitions and applied to a variety of papers reporting all types of procedures. This definition was continuously updated and applied iteratively to all articles. The definition has three key components: (1) method of access to the body, (2) instrumentation and (3) requirement for operator skill. It therefore encapsulates all types of invasive procedure regardless of the method of access to the body (incision, natural orifice or percutaneous access), and is relevant whatever the clinical discipline (eg, obstetric, cardiac, dental, interventional cardiology or radiology). Crucially, the definition excludes medicinal products, except where their administration occurs within an invasive procedure (and thereby involves operator skill). The application of a universal definition of an invasive procedure will (1) inform the selection of relevant methods for study design, (2) streamline evidence synthesis and (3) improve research tracking, helping to identify evidence gaps and direct research funds.

## INTRODUCTION

Invasive procedures, including surgery, are fundamental to healthcare. Worldwide, there are at least 230 million procedures performed annually and numbers are likely to increase due to the widening application of minimally invasive and image guided techniques.[1] Despite the volume of invasive procedures undertaken, the number and quality of randomised controlled trials (RCTs) in this area has historically been poor. Papers examining the quality of surgical RCTs have repeatedly demonstrated limitations in study design and conduct, such as recruitment, quality assurance of interventions and the blinding of trial personnel.[2–4] In the absence of evidence from well designed and conducted RCTs, clinical practice has been largely driven by personal preference, experience and anecdote. This results in variations and inequalities between surgeons, centres and regions with respect to the indications for, and types of, invasive procedures performed.[5–7]

### Cultivating research in invasive procedures

In the UK, the situation has begun to improve. The number and quality of funded RCTs in surgery is increasing, which has been facilitated by methodological advances and a marked shift in research culture. The Royal College of Surgeons of England has invested in surgical trials centres[8] and networks of research-active surgeons have been established.[9] These activities have resulted in ~50 new surgical RCTs in the last 5 years and over 150 new chief and principal investigators. As a result, the number of patients entering surgical RCTs has doubled.[10] Although these improvements have largely centred around surgery, the underlying principles are common to invasive procedures undertaken in other clinical disciplines such as cardiology, gastroenterology and radiology. To maximise the opportunities afforded by these initiatives, it is now necessary to understand exactly what is meant by an invasive procedure by developing a transparent and practical definition.

### Why is it important to define invasive procedures?

A clear definition of invasive procedures has several benefits. It would, (1) inform the selection of relevant methods for study design, (2) streamline evidence synthesis and (3) improve the accuracy of categorisation and tracking of research activity.

### Designing studies to evaluate invasive procedures

Evaluation of invasive procedures requires the application of specific methods to optimise trial design and conduct. These differ

**BMJ**

from those required in pharmaceutical studies. One main difference is that invasive procedures are complex interventions, with multiple interacting parts that can act independently or interdependently to influence outcomes.[11] Specific design features include the need for iterative development work in early phase studies before undertaking a main trial. This may involve establishing the parameters of intervention standardisation, methods for blinding trial personnel and participants, and assessing adherence to treatment protocols. Challenges during later phase studies (ie, RCTs) include recruitment and the need to account for operator skill and expertise at either the individual and/or centre level. These features are common to studies evaluating all types of invasive procedures, regardless of anatomical area or clinical discipline, and recognition of this would optimise study design and conduct, and clarify governance requirements.

### Streamlining evidence synthesis

Developing and applying a common definition for invasive procedures has the potential to make systematic literature searching more efficient and sensitive. This is especially relevant for reviews investigating groups of procedures. For example, a review synthesising evidence regarding surgical interventions for a particular condition may draw different conclusions depending on the definition of surgery used. Similar problems are apparent for methodological reviews investigating surgical procedures as a whole.

Another problem is that it is currently not possible to search for studies of invasive procedures without developing extensive keyword lists, because terms such as 'surgery' and 'invasive procedure' do not consistently identify relevant papers. Searches can then be difficult to reproduce because authors define surgery in different ways using different strategies and Medical Subject Headings (MeSH).[4 12 13] A common definition for invasive procedures linked to a working search strategy and MeSH term would facilitate these reviews by minimising the number of irrelevant papers retrieved and reducing the risk of missing relevant papers.

### Research tracking

Accurate tracking of research involving invasive procedures is vital for the strategic prioritisation of future RCTs. Tracking can help demonstrate output to funding bodies, identify evidence gaps, provide funds for under-researched areas and to reduce research waste. A common definition for invasive procedures would provide transparent information about research activities and promote the accurate categorisation of studies.

### Existing definitions

Currently, there is no widely accepted definition of an invasive procedure and the terms surgery and 'interventional procedure' are characterised inconsistently. Some definitions include only procedures that physically change the anatomy,[2] involve making a cut, are undertaken in a

sterile environment or use anaesthesia.[4] Each of these has limitations. For example, requiring that procedures physically change anatomy will exclude invasive diagnostic procedures (eg, laparoscopy, arthroscopy). Definitions specifying that procedures should involve a cut will miss those undertaken via natural orifices (eg, endoscopy) or using percutaneous techniques (eg, cardiac catheterisation), which are also invasive. The need for a sterile environment and/or anaesthetic would also potentially exclude these types of procedures from the definition.

Further definitions of surgery are based on the personnel involved in the study regardless of the nature of the intervention, such that any research involving surgeons is labelled surgical.[14] This poses problems as studies of pharmaceutical interventions delivered to surgical patients will be deemed 'surgical', whereas they actually require research methods and governance appropriate for the evaluation of pharmaceutical interventions rather than invasive surgical procedures.

### Proposal for a comprehensive definition of invasive procedures

We propose a definition for invasive procedures that addresses the limitations of those currently available. Our definition was developed from an analysis of the 3946 papers from the last decade. Initially, a preliminary definition was created based on existing definitions and applied to a variety of papers reporting all types of procedures. The preliminary definition was continuously updated and applied iteratively to all articles, thereby verifying that the final definition could be applied to the entire spectrum of invasive procedures (box 1). The definition has three key components: (1) method of access to the body, (2) instrumentation and (3) requirement for operator skill. This definition encapsulates all types of invasive procedure regardless of the method of access to the body (incision, natural orifice or percutaneous access) or clinical

---

**Box 1  Proposed definition of an invasive procedure**

► An invasive procedure is one where purposeful/deliberate access to the body is gained via an incision, percutaneous puncture, where instrumentation is used in addition to the puncture needle, or instrumentation via a natural orifice. It begins when entry to the body is gained and ends when the instrument is removed, and/or the skin is closed. Invasive procedures are performed by trained healthcare professionals using instruments, which include, but are not limited to, endoscopes, catheters, scalpels, scissors, devices and tubes.

► Where invasive procedures also involve the administration of a medicinal product, these could be categorised as being part of an 'invasive procedure' when operator skill is required for its administration within the body, that is, when an internal action is performed to administer the product or the product is administered to a targeted anatomical area, for example, Zhu *et al.*[15] There are also procedures which involve operator skill to target something inside the body (eg, electromagnetic radiation in the eye) without an incision, percutaneous puncture or instrumentation via a natural orifice. These types of procedures do not fall within the definition of an invasive procedure.

discipline (eg, obstetric, cardiac, dental, intervention radiology and so on). Crucially, the definition excludes medicinal products, except where their administration occurs within an invasive procedure (and thereby involves operator skill).

## Patient perspectives

Three patients who had previously undergone an invasive procedure provided feedback on the proposed definition of invasive procedures. The patients expressed that their view of invasive procedures was not centred on how access to the body was obtained, but rather that 'it's not about a cut, it's about something entering your body'. One patient stated, 'surgery is not all about cutting… I think that's quite an old-fashioned view. There are more procedures around now that may not involve cutting and a definition needs to include those'. Furthermore, the purpose of the invasive procedure, whether diagnostic or therapeutic, was not expressed as being an important criterion in whether a procedure is defined as invasive, and thus has not been included in the proposed definition.

## CONCLUSION

We propose a comprehensive way of defining invasive procedures. Agreeing and applying a definition to this fundamental aspect of healthcare is crucial, to optimise study design and conduct, facilitate evidence synthesis and improve the tracking of research activity.

**Acknowledgements** The authors wish to thank Alan Thomas, Azmina Verjee and Elizabeth Locke for their valued input as patient advisers.

**Contributors** All authors are based in the Bristol Centre for Surgical Research, the Surgical Innovation theme of the Bristol Biomedical Research Centre (BRC) and the MRC ConDuCT-II (Collaboration and innovation in Difficult and Complex randomised controlled Trials In Invasive procedures) Hub for Trials Methodology Research. SC is a research fellow and NSB and JMB are academic surgeons (MRC Clinician Scientist and professor of surgery, respectively). JMB is an NIHR senior investigator. Repeated challenges in designing and conducting methodological and applied research including recurrent requests from other research groups for advice in this area have led to conceptualisation and writing of this article. SC, NSB and JMB all contributed to the development of the manuscript and approved its final version. JMB is the guarantor.

**Funding** This study was supported by the MRC ConDuCT-II (Collaboration and innovation in Difficult and Complex randomised controlled Trials In Invasive procedures) Hub for Trials Methodology Research and the NIHR Biomedical Research Centre at University Hospitals Bristol NHS Foundation Trust and the University of Bristol. The views expressed in this publication are those of the author(s) and not necessarily those of the NHS, the National Institute for Health Research or the Department of Health and Social Care.

**Competing interests** None declared.

**Patient consent for publication** Not required.

**Provenance and peer review** Not commissioned; externally peer reviewed.

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
