## [Reviewer comments · BMJ Open]

ARTICLE DETAILS

TITLE (PROVISIONAL)	What is an invasive procedure? A definition to inform study design, evidence synthesis and research tracking
AUTHORS	Cousins, Sian; Blencowe, Natalie; Blazeby, Jane

VERSION 1 - REVIEW

REVIEWER	Professor S Enoch Director of Postgraduate Surgical Studies and Education Professor - Higher Surgical Education and Training Doctor Academy Group (Intl) Cardiff, United Kingdom
REVIEW RETURNED	11-Feb-2019

GENERAL COMMENTS	There is a need in the scientific literature and within the medical community for this clear definition. Therefore, the idea and the definition merits publication. However, the length of the manuscript is a concern. Since it is a communication article without extensive research or methodology, the idea should be articulated in a much more succinct and pithy manner. Aside from the box and references, the ideal length for an article of this nature should be in the region of 750-800 words. Please amend the manuscript to the above recommendation.
---

REVIEWER	Dirk Moore Rutgers University, USA
REVIEW RETURNED	20-Mar-2019

GENERAL COMMENTS	The authors present a new definition of an invasive procedure, and this could be of value in describing procedures when publishing results of a clinical study. However I am unclear on specifically what types of studies where uncertainty about the definition of a procedure is problematic, nor how this definition would clarify the study. Regarding evidence synthesis, perhaps in regards to a meta-analysis, the authors point out that key word lists may not correctly identify relevant studies. But it is unclear specifically how this new definition would help this. Furthermore, if one were to do a meta-analysis, the types of invasive procedures
---

	involved would, I expect, be clearly specified, and then the meta-analysis search would continue on that basis.
--	---

VERSION 1 – AUTHOR RESPONSE

Reviewers' comments to the authors:

Reviewer 1:

1. There is a need in the scientific literature and within the medical community for this clear definition. Therefore, the idea and the definition merits publication.

Reply: Thank you for this positive comment.

2. However, the length of the manuscript is a concern. Since it is a communication article without extensive research or methodology, the idea should be articulated in a much more succinct and pithy manner. Aside from the box and references, the ideal length for an article of this nature should be in the region of 750-800 words.

Please amend the manuscript to the above recommendation.

Reply: Thank you for this comment. We agree that the work presented is not suited to a lengthy article type. We have again reviewed the text to make sure that the idea and definition is presented as succinctly as possible and have reduced the word count. However, based on editorial advice and that the current word count (1219) is within the 2500-word guideline for communications articles, we feel that further cuts at this stage would be detrimental in providing a clear background, rationale and presentation of the proposed definition.

Reviewer: 2

1. The authors present a new definition of an invasive procedure, and this could be of value in describing procedures when publishing results of a clinical study. However I am unclear on specifically what types of studies where uncertainty about the definition of a procedure is problematic, nor how this definition would clarify the study.

Revision: We have now clarified in the text the types of studies where a common transparent definition would be most useful. Paragraph 2 on page 5 now reads –

“Developing and applying a common definition for invasive procedures has the potential to make systematic literature searching more efficient and sensitive. This is especially relevant for reviews investigating groups of procedures. For example, a review synthesising evidence regarding surgical interventions for a particular condition may draw different conclusions depending on the definition of surgery used. Similar problems are apparent for methodological reviews investigating surgical procedures as a whole.”

2. Regarding evidence synthesis, perhaps in regards to a meta-analysis, the authors point out that key word lists may not correctly identify relevant studies. But it is unclear specifically how this new definition would help this. Furthermore, if one were to do a meta-analysis, the types of invasive procedures involved would, I expect, be clearly specified, and then the meta-analysis search would continue on that basis.

Reply: Many thanks for this helpful comment. We have revised the manuscript to clarify this. Systematic reviews, including meta-analyses, would benefit from a clear definition of 'invasive procedures'. Whilst this may be less relevant if one procedure is being compared with another, it may be particularly helpful if the review investigates a group of procedures. For example, a review synthesising evidence regarding surgical interventions for a particular condition may draw different conclusions depending on the definition of surgery used, which is currently heterogeneous.

Similar problems are apparent for methodological reviews investigating surgical procedures as a whole. For example, a recent systematic review¹ investigating the methodological aspects of placebo-controlled trials of surgery identified 63 trials, of which one third were endoscopic interventions. If this review had used a definition of surgery including only on procedures where a cut was made, numerous relevant studies would have been missed.

Revision: We have now clarified that in the text on page 5, paragraph 2, which now reads.

“Developing and applying a common definition for invasive procedures has the potential to make systematic literature searching more efficient and sensitive. This is especially relevant for reviews investigating groups of procedures. For example, a review synthesising evidence regarding surgical interventions for a particular condition may draw different conclusions depending on the definition of surgery used. Similar problems are apparent for methodological reviews investigating surgical procedures as a whole.”